# The NAMPT Inhibitor FK866 in Combination with Cisplatin Reduces Cholangiocarcinoma Cells Growth

**DOI:** 10.3390/cells12050775

**Published:** 2023-02-28

**Authors:** Kishor Pant, Seth Richard, Estanislao Peixoto, Jun Yin, Davis M. Seelig, Pietro Carotenuto, Massimiliano Salati, Brunella Franco, Lewis R. Roberts, Sergio A. Gradilone

**Affiliations:** 1The Hormel Institute, University of Minnesota, Austin, MN 55912, USA; 2Masonic Cancer Center, University of Minnesota, Minneapolis, MN 55455, USA; 3Division of Clinical Trials and Biostatistics, Mayo Clinic, Rochester, MN 55905, USA; 4Comparative Pathology Shared Resource, Masonic Cancer Center, College of Veterinary Medicine, University of Minnesota, St. Paul, MN 55108, USA; 5Telethon Institute of Genetics and Medicine (TIGEM), 80078 Pozzuoli, Italy; 6Medical Genetics, Department of Translational Medical Sciences, University of Naples Federico II, 80131 Naples, Italy; 7Medical Oncology Unit, University Hospital of Modena, 41125 Modena, Italy; 8Clinical and Experimental Medicine, University of Modena and Reggio Emilia, 411250 Modena, Italy; 9Genomics and Experimental Medicine Program, Scuola Superiore Meridionale, School for Advanced Studies, 80131 Naples, Italy; 10Department of Gastroenterology and Hepatology, Mayo Clinic, Rochester, MN 55905, USA

**Keywords:** NAMPT, NAD, FK866, Cholangiocarcinoma, cisplatin

## Abstract

It is well established that Cholangiocarcioma (CCA) drug resistance plays a crucial role in the spread and survival of cancer cells. The major enzyme in the nicotinamide-adenine dinucleotide (NAD+)-mediated pathways, nicotinamide phosphoribosyltransferase (NAMPT), is essential for cancer cell survival and metastasis. Previous research has shown that the targeted NAMPT inhibitor FK866 reduces cancer cell viability and triggers cancer cell death; however, whether FK866 affects CCA cell survival has not been addressed before. We show herein that NAMPT is expressed in CCA cells, and FK866 suppresses the capacity of CCA cells to grow in a dose-dependent manner. Furthermore, by preventing NAMPT activity, FK866 significantly reduced the amount of NAD+ and adenosine 5′-triphosphate (ATP) in HuCCT1, KMCH, and EGI cells. The present study’s findings further show that FK866 causes changes in mitochondrial metabolism in CCA cells. Additionally, FK866 enhances the anticancer effects of cisplatin in vitro. Taken together, the results of the current study suggest that the NAMPT/NAD+ pathway may be a possible therapeutic target for CCA, and FK866 may be a useful medication targeting CCA in combination with cisplatin.

## 1. Introduction

Due to the poor prognosis and resistance to chemotherapies, treating patients with Cholangiocarcinoma (CCA) is difficult [1,2]. Given the genetic variability both in tumors and intratumor, chemoresistance is extremely difficult to overcome. These variables can make patients fully refractory to chemotherapy, and in those who do react, the tumor’s selection of resistant clones can lead to cancer recurrence. Cancerous cells modify their metabolism to survive and flourish in the hostile tumor microenvironment due to hypoxic circumstances and limited availability to nutrients [3]. One of the mechanisms that is activated is the Warburg effect, which results in increased glycolysis and excessive lactate generation. This process, sometimes referred to as aerobic glycolysis, promotes the growth of tumors [1,4]. The high rates of aerobic glycolysis change the NADH/NAD+ redox ratio by interfering with NAD+ metabolism. Instead of transferring electrons from NADH to the mitochondrial respiratory chain, pyruvate is converted to lactate in order to replenish cytosolic NAD+. The coenzyme called NAD+ is involved in energy transmission and nearly all metabolic activities in the cells. Among other biological processes, it controls transcription, cell cycle progression, apoptosis, DNA repair, and metabolic regulation [5,6].

It has been determined that the cellular metabolism is directly influenced by the homeostatic mechanism of cellular NAD+ levels. The quantity is maintained constant by a closely regulated balance between NAD+ synthesis, NAD+ breakdown, and NAD+ recycling via a salvage mechanism [6]. Cancer cells heavily rely on the NAD+ salvage pathway for their metabolic processes. This process is heavily dependent on the enzymes nicotinamide phosphoribosyl transferase (NAMPT) and nicotinamide mononucleotide adenylyl transferase (NMNAT1) [7]. The most active enzyme is NAMPT, which catalyzes the rate-limiting step of nicotinamide condensation with 5-phosphoribosyl-1-pyrophosphate to form nicotinamide mononucleotide. NMNAT1, which makes NAD+ from nicotinamide mononucleotide and ATP, is the next enzyme in line. NAMPT has a function in tumor cell survival and growth and is upregulated in a number of solid tumors [7].

A potent anticancer drug in both in vitro and in vivo settings, FK866 is a highly selective noncompetitive NAMPT inhibitor [8]. Numerous recent investigations have demonstrated that it prevents the growth of numerous cancer cell types while having no effect on healthy cells [8,9,10]. The effects of NAMPT inhibition in CCA have not yet been addressed. Therefore, we sought to determine whether the combinatorial targeting of NAMPT with standard chemotherapy shows promise to target CCA cells. In HuCCT1, KMCH, and EGI, the NAMPT inhibitor FK866 was examined to determine if it may enhance the effects of the regularly used chemotherapy drug cisplatin.

## 2. Materials and Methods

### 2.1. Cell Culture, Treatments, and Transfections

HuCCT1, EGI, and KMCH CCA cells were grown in DMEM (Thermo Fisher Scientific, Waltham, MA, USA) containing 10% FBS and 1% penicillin/streptomycin. Additionally, normal human cholangiocyte cell lines (H69 and NHC) were employed in this work. 0.2–50 nm of FK866 (Tocris Bioscience, Bristol, UK) and 0.2–20 μM of Cisplatin (Tocris Bioscience, Bristol, UK) were applied to the cells at different concentrations for 2–3 days. Lipofectamine RNAiMAX (Thermo Fisher Scientific) was used to transfect NAMPT-siRNA (10 nM) (Dharmacon Inc., Lafayette, CO, USA) and non-Target-siRNA (NT) in the HuCCT1 cells.

### 2.2. Patients’ Samples and TMA

A total of 175 Tissue Microarrays (TMA) CCA patient human samples were examined for the investigations involving human participants at the Mayo Clinic’s Hepatobiliary SPORE in Rochester, MN, USA. Two tissue core samples from each patient’s tumor and unrelated tissue were used to create a TMA, which was then immunohistochemically examined for NAMPT expression (Proteintech, Rosemont, IL, USA). Not all patients gave written informed permission for the TMAs. The Mayo Clinic Institutional Review Board has previously authorized the inclusion of both patients who provided written informed permission and dead patients who did not decline study, as required by Minnesota law, in our Hepatobiliary Neoplasia Biorepository (IRB 707-03). Paraffin blocks from these individuals kept in the Mayo Pathology Tissue Archives were used to build TMAs [11].

### 2.3. Microdissection and RNA-Seq Analysis

Prior to microdissection, a pathologist identified the tumor stroma in sampled tissue. Thermo Scientific’s ArcturusXTTM Laser Capture Microdissection System and Life Technologies’ Arcturus^®^ CapSure^®^ Macro LCM Caps were utilized for microdissection. According to the manufacturer’s instructions, highly enriched populations of normal or tumor-associated stroma from the samples were found and separated. The same slides that contained the tumor (T) also contained samples of normal (N), adjacent (A), and tumor-associated stroma (S).

Furthermore, Total RNA was isolated from the FFPE (Formalin-Fixed-Paraffin-Embedded) samples from 47 matched tumors from CCA patients who underwent a retrospective evaluation to determine if they had bile duct cancer from September 2005 to January 2020. Interest-generating clinicopathologic, biochemical, and pretreatment demographic characteristics were obtained. The Ethical Committee of the University Hospital of Modena in Italy gave its approval for the use of human samples. At the University Hospital of Modena in Italy, each patient completed a general informed permission form authorizing the use of the data for research purposes and analysis.

### 2.4. Western Blot Analysis

By collecting cells with RIPA buffer and a protease inhibitor cocktail, total cellular protein was isolated. The gels were loaded with an equal quantity of protein (around 10 μg/well), which was then separated in 10% SDS-PAGE before being transferred to nitrocellulose membranes. The membranes were blocked with BSA (5%) before being treated with primary antibodies for 12 h at 4 °C. They were further treated with the appropriate secondary antibodies after being rinsed with Tris-buffered saline +0.05% Tween 20 (TBS-T). Thermo’s ECL Western blotting substrate (used to detect chemiluminescence) was used to generate signals after the membranes had been washed.

### 2.5. Oxygen Consumption Rates (OCR) Analysis

To form a cell monolayer, cells were seeded into an XF96-well cell culture plate at a density of 2 × 105 cells per well. The next day, the cell media were changed to XF Base medium (pH 7.4) with 2 mM pyruvate and 20 mM glucose as supplements. Using the XF-96 Analyzer, oxygen consumption rates (OCR) were used to directly assess mitochondrial respiration (Seahorse Bioscience, Chicopee, MA, USA).

### 2.6. ATP Assay

RealTime-Glo™ ATP Assay kit from Promega was used. ATP assay was measured in cultured cells as per manufacturer’s protocols.

### 2.7. IncuCyte Cell Growth Analysis

Before the FK866 and cisplatin treatments, 96-well plates with approximately 1000 cells per well were incubated for 12 h. For the purpose of analyzing cell growth, all the cells were cultured in IncuCyte for 72 h.

### 2.8. MTS Assay

According to the manufacturer’s instructions, the MTS test was carried out following FK866 and NMN treatments for 48 h to detect cell proliferation (Promega, WI, USA).

### 2.9. 3D Spheroids Formation Assay

A total of 2000 cells were plated on 96-well Corning round-bottom plates and centrifuged for 3 min at 1500 rpm. The IncuCyte live-cell analysis system was used to incubate the plate, and a repeat scan with a 10× objective was scheduled for every four hours. After each photograph, the spheroid area was estimated and displayed as a bar graph [12].

### 2.10. ROS Assay

ROS measurements were performed using the ROS-Glo H2O2 assay (Promega) according to the manufacturer’s instructions. In 96-well plates, cells were seeded in opaque, white plates. Luminescence was evaluated with an Iluminometer. Standard deviations from triplicate samples (SD) and average relative light units (RLU) were calculated. FK866 (10 nM) and 500 uM N-acetylcysteine (NAC) were used as test compound and negative controls, respectively.

### 2.11. Immunohistochemistry

In order to prepare TMA tissue for immunohistochemistry (IHC), 4 µm formalin-fixed, paraffin-embedded sections were deparaffinized and rehydrated, followed by antigen retrieval using 10 mM citrate buffer pH 6.0 in a steamer, all of which was performed before IHC operations using a Dako Autostainer. A rabbit polyclonal antibody from Proteintech was used as the main antibody in IHC for NAMPT. With DAB as the chromogen, the rabbit EnVision+ Kit (Dako catalog K4011) allowed for the detection of a bound primary antibody. TMA IHC preparations were examined under a light microscope, and tumor cell cytoplasm and/or nuclei, as well as the normal bile duct that surrounds them, were positively labeled. This assessment was conducted using a semi quantitative 0–4 scale, where 0 is absent, 1 is minimal, 2 is mild, 3 is moderate, and 4 is marked.

### 2.12. Immunofluorescence

HuCCT1 and NHC cells were cultured on glass slides for three days. After fixation and blocking, the primary antibody against NAMPT (ProteinTech, Rosemont, IL, USA) was used at 4 °C overnight. For fluorescent detection, a goat anti-mouse secondary antibody labeled with Alexa Fluor dye (Invitrogen/Life Technologies) was incubated for 1 h. DAPI was used to label the nuclei. A Zeiss Apotome microscope was used to examine NAMPT expression.

### 2.13. Statistics Analysis

All experiments were performed at least three times (n = 3). The data were expressed as mean ± standard deviation. Unpaired Student’s *t*-test between two groups and one-way ANOVA were performed among the groups. The data were considered statistically significant if *p* < 0.05.

## 3. Results

### 3.1. NAMPT Expression in CCA Cells

In order to check the expression of NAMPT in CCA, first we performed IHC in human samples. The tumor population showed intracytoplasmic NAMPT expression in all samples (Figure 1A). The intensity of this staining varied but there were no significant differences with the normal samples (Figure 1B). There was often heterogeneity in the intensity of NAMPT staining within a given sample (e.g., areas of Grade 1 staining adjacent to areas of Grade 3 staining). In the assignment of an intensity score for these samples, the dominant staining intensity was chosen. Although many of the tumor samples demonstrated moderate to markedly intense NAMPT staining, there was a subset of cases that demonstrated notably less intense staining; however, no significant differences in patient survival were noted (Appendix A and Appendix A). Similarly, we observed no change in the NAMPT expression in the Italian CCA patients’ tumor samples (T) compared to the normal and stromal area (Figure 1C). We then performed Western blotting analysis in CCA and normal cholangiocyte cell lines. We found that NAMPT expression was significantly increased in the CCA cells (HuCCT1 and EGI) compared to the normal cholangioytes (NHC and H69) (Figure 1D,E). These data indicate that CCA tumors from patients express NAMPT and that in in vitro cell models, NAMPT expression is increased in CCA cells. Finally, using NAMPT immunofluorescence analysis, we found that NAMPT is also expressed both in the nucleus and cytoplasm of NHC and HuCCT1 cells (Figure 1F).

### 3.2. FK866-Mediated NAMPT Inhibition-Modulated CCA Cell Growth

Next, we checked effects of NAMPT inhibition on normal cholangiocytes, NHC, and CCA cell growth and found that FK866, a specific inhibitor of NAMPT, decreased CCA cell (HuCCT1, KMCH, and EGI) growth in a concentration (1–20 nM)-dependent manner, while minimally affecting normal cells (Figure 2A–D). Similarly, we also noticed that treatment with FK866 (1–20 nM) significantly decreased the size of the spheroid in CCA cells (HuCCT1 and KMCH), but a very small effect in the normal NHC cells was observed. (Figure 2E–H). Furthermore, to further assess NAMPT’s role in proliferation, we tested an siRNA-mediated reduction in NAMPT (Appendix A) in HuCCT1 cells and observed a significantly inhibited cellular proliferation (Appendix A).

### 3.3. Restoration of NAD^+^ Level Reverse the FK866 Effects in the Cells

As shown above, treatment with FK866 (10 nM) reduced CCA cell growth. However, pretreatment with nicotinamide mononucleotide (NMN, the product of NAMPT activity) significantly restored the cell growth in the CCA cells (Figure 3A–C). In the same way, treatment with FK866 in HUCCT1 and KMCH cells significantly decreased colony formation, and this effect was reversed with NMN treatment (Figure 3D–G). In order to substantiate the outcomes of our study, we performed a spheroid formation assay. Consistently, we found that FK866 (10 nM) treatment significantly reduced spheroid area in HuCCT1 and KMCH cells, and spheroid area was found to be increased with NMN treatment (Figure 3H–J).

### 3.4. NAMPT Inhibition Disrupts Mitochondrial Function in CCA Cells

Next, we investigated the mitochondrial function in the presence of FK866 (10 nM) and we found a low oxygen consumption rate (OCR) in HUCCT1 and KMCH cells. Treatment with NMN, however, significantly restored the OCR level in these cells (Figure 4A–C). Subsequently, we noticed that the total cellular ATP level in the HUCCT1, KMCH, and EGI was lowered in the presence of FK866 treatment, an effect that was reversed by NMN addition (Figure 4C–E). It is evident that mitochondria dysfunction leads to an augmented level of reactive oxygen species (ROS) [13]; thus, we checked the ROS level in HuCCT1, KMCH, and EGI cells in the presence of FK866 and found an elevated level of ROS in these cells with FK866 treatment; however, ROS levels were reduced with the addition of an antioxidant, NAC, or NMN (Figure 4F–H). These data indicate that FK866-mediated NAMPT inhibition affects mitochondrial function in CCA cells. Finally, since NAMPT expression is known to influence cytokine levels, we assessed the expression of TGF β2 and IL6 after FK866 treatment and found suppressed expression in HuCCT1 cells (Appendix A).

### 3.5. FK866 Potentiates the Effects of Cisplatin in CCA Cells

Finally, we checked the effect of the anticancer drug, cisplatin, in combination with NAMPT inhibition by FK866. We found that FK866 (0.2–1 nm) significantly inhibited cell proliferation in HuCCT1, KMCH, and EGI cells in the presence of cisplatin (0.2–1 μM) compared to their treatments alone (Figure 5A–C). These data indicate that cisplatin at lower doses could be effective in combination with FK866 in CCA cells.

## 4. Discussion

CCA is one of the most frequent forms of malignant tumors, with a poor prognosis and a frequent clinical recurrence rate in patients. This may be partly because CCA cells have high rates of drug resistance and paucity of therapeutic options [1,14]. Drug resistance plays a critical role in the processes of CCA cell survival, and is the most clinically significant feature of these tumor [15,16]. Therefore, it is imperative to find possible innovative targeted medicines that are efficient against CCA. We described here that NAMPT is expressed in CCA, and FK866, a new small-molecule NAMPT inhibitor, may be a potential targeted medication for the treatment of patients with CCA as it was found in the current investigation to block NAMPT-mediated NAD synthesis, affecting cell metabolism and proliferation preferentially in tumor cells compared to normal cholangiocytes.

The NAMPT/NAD synthesis pathway engages in a number of vital biological activities in both healthy and malignant cells, including transcription, cell cycle progression, DNA repair, circadian rhythm, and metabolic control [7,13]. The salvage process is mostly used to produce NAD+. The conversion of nicotinamide, one of the primary precursors of NAD+, into nicotinamide mononucleotide is catalyzed by NAMPT, which is the rate-limiting enzyme in the NAD+ salvage route. In comparison to normal cells, tumor cells require higher NAD+ for cell growth [7]. The results of the current investigation confirmed earlier studies in others types of tumors including pancreatic, gastric, and leukemia, showing that suppressing NAMPT activity lowers cell proliferation [9,17,18,19]. For example, pancreatic adenocarcinoma cells are more susceptible to the tumor-specific, PARP-independent metabolic catastrophe and cell death caused by β-lapachone when NAMPT is inhibited [17]. However, in five clinical investigations using three specific NAMPT inhibitors and a total of 104 patients, no meaningful tumor remission was identified (FK866, CHS828, and GMZ1777) [18,19,20]. FK866’s short half-life in circulation demands prolonged treatment regimens, which harm developing hematopoietic cells, according to clinical research. NAD+-depleting medications, such as NAMPT inhibitors, are therefore likely to be unsuccessful when used alone due to low tumor selectivity [21,22,23]. FK866 has, thus, been researched as an addition to other well-known chemotherapies. It enhanced the 5-fluorouracil (5FU) chemosensitivity of gastric cancer cells [19,24] and enhanced the effects of cisplatin and etoposide on neuroblastoma cell lines, and dramatically reduced the xenografts’ overall metabolic activity, which impacted the survival of cancer cells [25]. Combining drugs has been shown to be a successful strategy for treating cancers such as pancreatic ductal adenocarcinoma (PDAC) that are resistant to chemotherapy. This medicine combination allowed for lower dosages to be administered with fewer side effects while still promoting cell death and reducing drug resistance; consistently, our data showed that FK866 allows the inhibition of CCA cell proliferation with lower cisplatin doses that are harmless for the normal cholangiocytes cell lines. 

Another effect of NAMPT inhibition is the decreased activity of SIRT1, a deacetylase that we previously showed is overexpressed in CCA and a potential target for its treatment [11,12]. Consistently, NAMPT and SIRT1 levels have been seen to rise in HCC and other cancers, a condition linked to a bad prognosis for those who have the disease [10,26,27,28]. Additionally, SIRT1 plays crucial roles in HCC stem cell self-renewal and fosters tumor cell invasiveness [26]. Cancer cell growth is primarily facilitated by the EMT marker proteins e-cadherin and vimentin, and SIRT1 is connected to the production of invasive proteins in tumors [29,30]. NAMPT plays a key role in the development of cancer by promoting EMT in numerous cancers [19]. Our results suggests that NAMPT has an impact on the cell growth and survival of CCA cells, but the specific molecular mechanism by which this occurs has not yet been identified. 

Further research is, therefore, necessary to understand the inhibitory control of the NAMPT/NAD+ pathway. As a NAMPT inhibitor, FK866 dramatically reduces NAD+ levels by blocking NAMPT. The present study’s findings further demonstrate that FK866 lowers NAD+ and ATP levels and reduces the survival of CCA cells by preventing NAMPT activity in HuCCT1, KMCH, and EGI cells.

According to another study, FK866 offers a unique treatment approach to increase the effectiveness of chemotherapy drugs such as etoposide against leukemia [18]. Additionally, FK866 dramatically improves the anticancer activity of gemcitabine in pancreatic ductal adenocarcinoma cells and orthotopic xenograft animal models, suggesting that FK866 may help in the treatment of pancreatic cancer by lowering NAD+ levels and reducing glycolytic activity [31]. FK866 does reduce tumor invasion and metastasis; however, this is only occasionally documented. The NAMPT-specific inhibitor FK866’s inhibitory role in CCA was revealed in the current work for the first time, suggesting that FK866 may be employed as a new anti-CCA drug. 

According to the current study’s findings, FK866 may inhibit the NAMPT signaling pathway in CCA cells, which lowers the levels of NAD+ and ATP and prevents mitochondrial metabolism. The NAMPT/NAD+ pathway may be a possible therapeutic option for treating CCA cell growth alone or in combination with cisplatin, according to the results of this study, which overall point to FK866 as a potentially successful targeted CCA treatment. Importantly, the combined use of FK866 with cisplatin allowed us to use reduced concentrations of chemotherapy in vitro, which showed efficacy on tumor cell lines but no significant effects on the normal cholangiocytes.

The expression levels of NAMPT in the human samples did not show significant changes in CCA samples at the protein level; however, a small but significant difference at the mRNA level was found in a different cohort. In contrast, the CCA cell lines analyzed showed increased NAPT levels compared to normal controls. Perhaps culture conditions and adaptation to in vitro growth conditions may explain the differences. Nevertheless, the apparent increased dependence of tumor cells on NAMPT activity still makes this enzyme an attractive target to explore. Therefore, further human-derived organoids, PDXs, and/or animal CCA models are warranted in future studies.

## 5. Conclusions

In conclusion, a variety of downstream protein regulatory pathways that collectively make up a systemic regulatory network in CCA cell growth and invasion are linked to energy metabolism through NAMPT/NAD+ and are the potential targets for cancer therapeutics.

## Figures and Tables

**Figure 1 cells-12-00775-f001:**
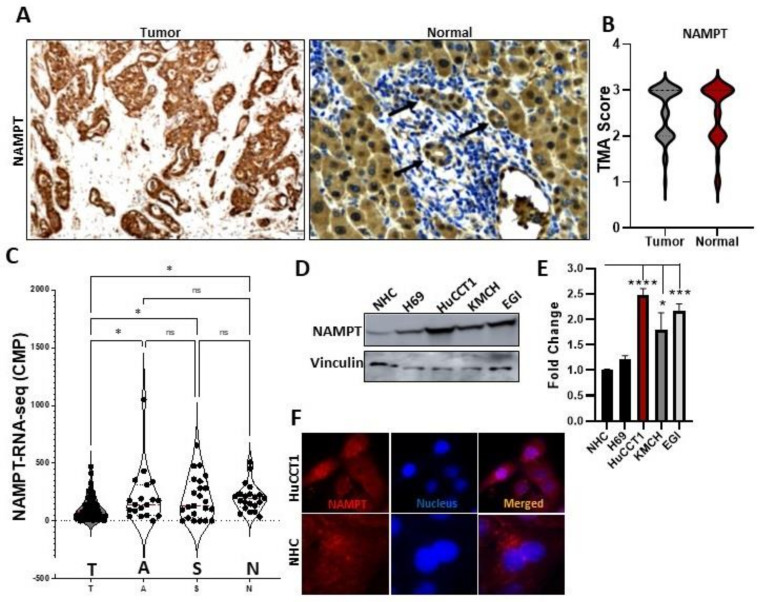
NAMPT expression in CCA cells. (**A**) On TMA slides, IHC for NAMPT in the CCA tumor was compared with the surrounding healthy tissue. (**B**) IHC scoring was performed for assessing NAMPT expression in tumors and normal bile duct. (**C**) RNA-seq analysis of the NAMPT gene in an independent cohort of CCA patients, where **T**—Tumor, **S**—surrounding stroma, and **N**—normal area. (**D**,**E**) NAMPT expression was analyzed in NHC, H69, HuCCT1, KMCH, and EGI cells with Western blotting technique and densitometry analysis was performed. (**F**) NAMPT immunofluorescence was conducted in the NHC and HuCCT1 cells (* *p*< 0.05, *** *p* < 0.001, **** *p* < 0.0001 and ns, non-significant).

**Figure 2 cells-12-00775-f002:**
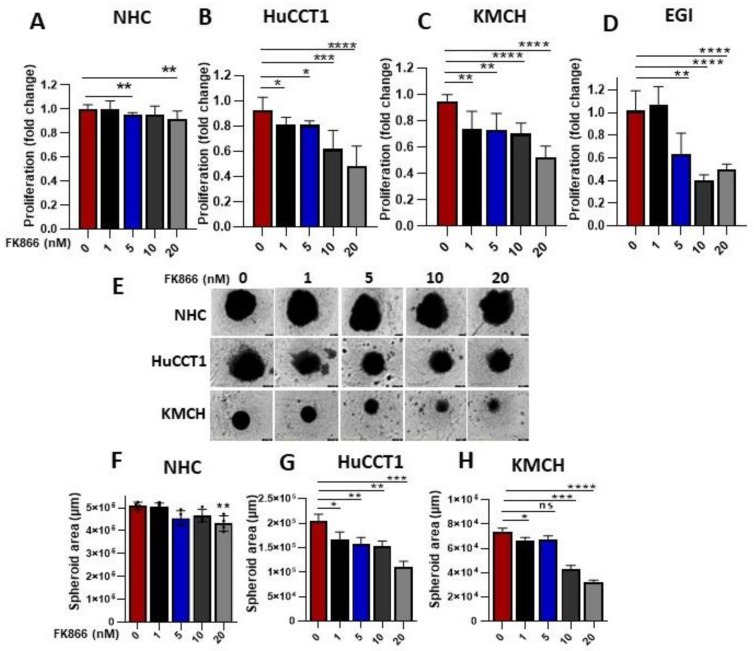
NAMPT regulates CCA cell growth. (**A**) FK866 (1–20 nM) was treated in NHC cells, and an MTS cell growth assay was performed. (**B**) HuCCT1 cells were treated with FK866 (1–20 nM) and cell growth assay was performed. (**C**) MTS assay was conducted in KMCH cells after FK866 treatment at different concentrations (1–20 nM). (**D**) Cell growth analysis was performed in EGI cells followed by the FK866 treatments. (**E**) Spheroid assay was performed in NHC, HUCCT1, and KMCH cell lines in presence of FK866 (0–20 nM). Images were taken in IncuCyte. (**F**–**H**)**,** Spheroid area was measured in NHC, HuCCT1, and KMCH cells after spheroid imaging by IncuCyte. (* *p* < 0.05, ** *p* < 0.01, *** *p* < 0.001, **** *p* < 0.0001, and ns, non-significant).

**Figure 3 cells-12-00775-f003:**
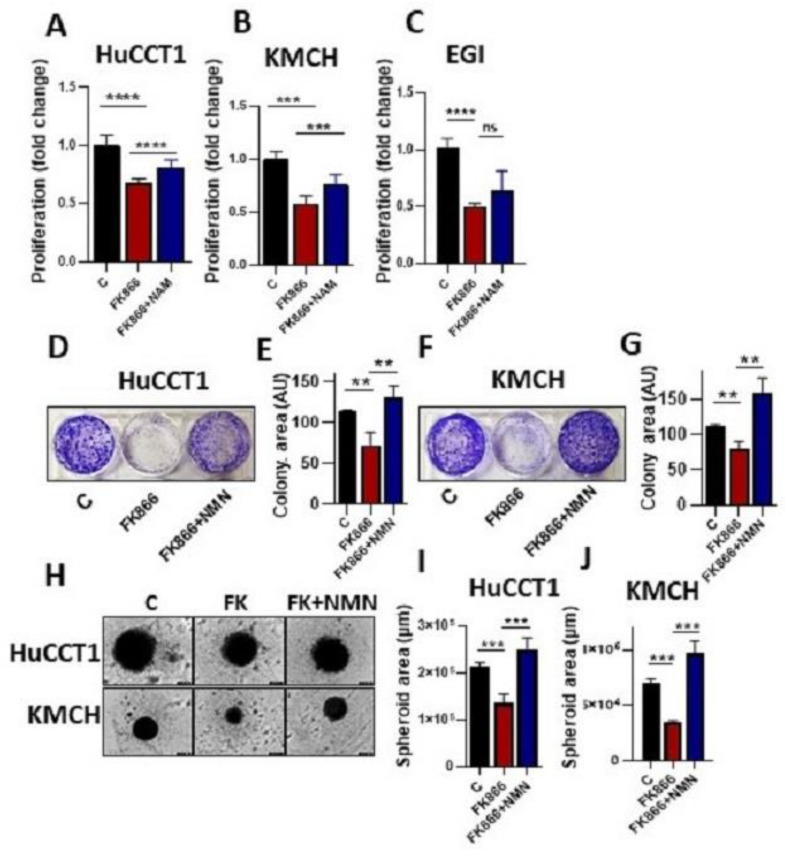
Supplementation with NAD reverses the effects of the FK866. (**A**–**C**) HuCCT1, KMCH, and EGI cells were treated with FK866 (10 nM) with or without NMN (10 μM), and cell growth assay was performed using MTS assays. (**D**–**G**) Colony formation assay was conducted in HuCCT1 and KMCH cells treated with FK866 for 10 days with or without NMN treatments. Colony area was measured by ImageJ analysis and represented as bar graphs. (**H**–**J**) Spheroid formation assay was performed in CCA cells (HUCCT1 and KMCH) in the presence of FK866 (10 nM) with or without NMN (10 μM). Images were captured and analyzed by IncuCyte. (** *p* < 0.01, *** *p* < 0.001, **** *p* < 0.0001 and ns, non-significant).

**Figure 4 cells-12-00775-f004:**
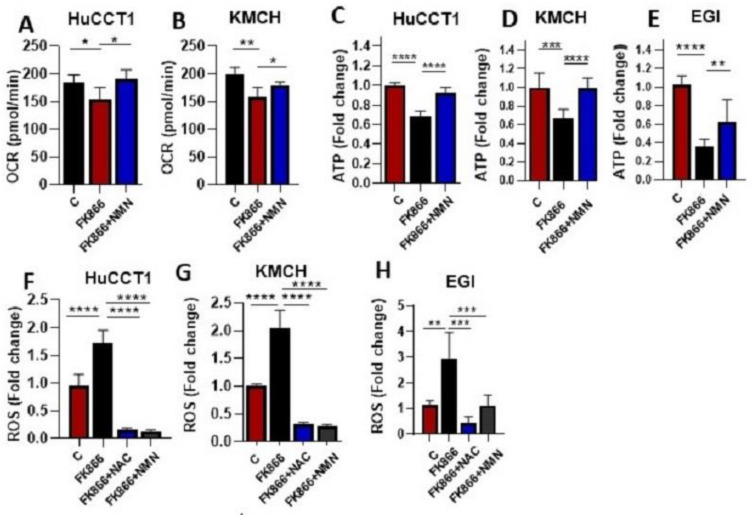
NAMPT inhibition disrupts mitochondrial metabolism. *(***A**,**B**) Seahorse metabolic phenotype assay was used to analyze the OCR values in CCA cells treated with FK866 with or without NMN for 48 h. (**C**–**E**) Total cellular ATP was analyzed in FK866 with or without NMN-treated CCA cells (HuCCT1, KMCH, and EGI). (**F**–**H**) Total cellular ROS was analyzed in FK866-treated HUCCT1, KMCH, and EGI cells with or without N-Acetyl cysteine (NAC) or NMN. ROS level is represented as relative fold change in bar graphs. (* *p* < 0.05, ** *p* < 0.01, *** *p* < 0.001, **** *p* < 0.0001).

**Figure 5 cells-12-00775-f005:**
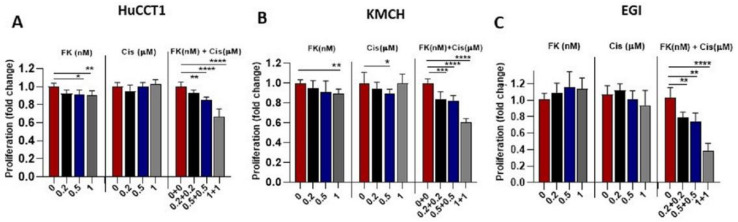
FK866 induces the effects of cisplatin in CCA cells. (**A**–**C**) CCA cells (HuCCT1, KMCH, and EGI) were treated with FK866 (0–1 nM), cisplatin (0–1 μM), or FK866 (0–1 nM) in combination with cisplatin (0–1 μM). MTS cell growth assay was performed and represented as relative fold change. (* *p* < 0.05, ** *p* < 0.01, *** *p* < 0.001, **** *p* < 0.0001).

## Data Availability

Not applicable.

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
