# Peer review of "The NAMPT Inhibitor FK866 in Combination with Cisplatin Reduces Cholangiocarcinoma Cells Growth"

_cells, 2023, doi:10.3390/cells12050775_

Round 1
Reviewer 1 Report
In this manuscript, the authors have explored the role of NAMPT in CCA. They discovered that FK866, a NAMPT inhibitor, could inhibit CCA cell growth and impair mitochondrial metabolism, In addition, the authors demonstrated that the combinational treatment of FK866 and cisplatin exhibited synergistic effect. Overall, the authors concluded that NAMPT could be a potential therapeutical target for CCA treatment. However, there are some remaining questions to be answered:
1, IHC scoring for NAMPT expression and RNA-seq analysis indicated the expression of NAMPT was not overexpressed or decreased in CCA patients. However, NAMPT overexpression was detected in three CCA cell lines. Could the authors explain the reasons?
2, The authors mentioned intracytoplasmic NAMPT expression was observed in all patients’ samples. Have the authors performed cell fractionation experiments to determine the findings in cell lines?
3, Have authors evaluate the cytotoxic effects of FK866 (1-20 nM) in normal cell lines? Does it affect proliferation on normal cell lines?
4, Have the authors performed knockdown or knockout experiments to validate NAMPT function?
5, Besides regulating the NAD+ pathway, does ROS also contribute to the effect of FK866? Could the inhibition effect caused by FK866 was rescued with NAC treatment?
6, For the combinational experiment, have the authors examined gemcitabine as well?
Author Response
Response to Reviewers Comments.
We really appreciate the thoughtful reviewers’ comments and that the manuscript was considered of interest. We have addressed the point-by-point comments bellow and made changes to the manuscript, accordingly, including new data (Figures 1F, 2A, Supp 3, and Supp 4).
Reviewer 1
In this manuscript, the authors have explored the role of NAMPT in CCA. They discovered that FK866, a NAMPT inhibitor, could inhibit CCA cell growth and impair mitochondrial metabolism, In addition, the authors demonstrated that the combinational treatment of FK866 and cisplatin exhibited synergistic effect. Overall, the authors concluded that NAMPT could be a potential therapeutical target for CCA treatment. However, there are some remaining questions to be answered:
Comment: IHC scoring for NAMPT expression and RNA-seq analysis indicated the expression of NAMPT was not overexpressed or decreased in CCA patients. However, NAMPT overexpression was detected in three CCA cell lines. Could the authors explain the reasons?
Response: We agree with the reviewer that there is a surprising difference between human samples and cell lines that are hard to explain. As we know, cancer is a heterogeneous condition with several genetic variants. Furthermore, variations in nutrients and growth factors in patient’s samples and in cultured cells may affect the redox enzymes including NAMPT. However, it is not obvious what the reasons could be. Perhaps culture conditions and adaptation to in vitro growth conditions may explain the differences. We have now added a paragraph in the discussion acknowledging these differences.
Comment: The authors mentioned intracytoplasmic NAMPT expression was observed in all patients’ samples. Have the authors performed cell fractionation experiments to determine the findings in cell lines?
Response: We highly appreciate the reviewer’s comment. We did not perform subcellular fractionations in this study, but we now performed immunofluorescence analysis of NAMPT in cell lines to verify cellular localization, and the observation is consistent with was previously reported by the pathologist in the human samples (New Figure 1F).
Comment: Have authors evaluate the cytotoxic effects of FK866 (1-20 nM) in normal cell lines? Does it affect proliferation on normal cell lines?
Response: Thank you so much for the suggestion. We now evaluated the effect of the NAMPT in normal Human cholangiocyte (NHC) and noticed that these cells were less sensitive to the treatment compared to CCA cells (New Figure 2A).
Comment: Have the authors performed knockdown or knockout experiments to validate NAMPT function?
Response: This is another excellent point, and it is really appreciated. We have now checked the effect of NMAPT si-RNA in CCA cells on NMAPT expression and cell proliferation (New Supp Figure 3).
Comment: Besides regulating the NAD+ pathway, does ROS also contribute to the effect of FK866? Could the inhibition effect caused by FK866 was rescued with NAC treatment?
Response: We thank reviewer for this valuable comment. In addition to acting as co-factor, NAD+ is a reducing agent which also plays an important role in redox equilibrium via directly inhibiting the ROS production in cells. We used NAC as a positive control along with our NMN treatment in our experiments. NAC inhibited ROS level in the cells when treated with FK866, and we think ROS also contribute to FK866 effect (Figure 4).
Comment: For the combinational experiment, have the authors examined gemcitabine as well?
Response: We acknowledge the reviewer’s comments. This time we only tested the effect of Cisplatin in combination with FK866. In future studies, we are also planning to check the effects of gemcitabine and other anticancer drugs used for CCA chemotherapy along with NAMPT inhibitors.

Reviewer 2 Report
In this article the authors tested the efficacy of the known NAMPT inhibitor, FK866 in reducing the survival of CCA cells. They demonstrated that by preventing the activity of NAMPT, the FK866 inhibitor not only suppresses the growth capacity of cancer cells, but also causes changes in mitochondrial metabolism. Furthermore, the experimental results show that a combination of drugs can be a successful strategy for the treatment of tumors resistant to chemotherapy alone. The conclusions are supported by the results and the results obtained are strong. The study has important clinical implications.
Comments to improve the study:
Since several polymorphisms have been found in NAMPT that could cause resistance to treatment, sequencing of RNA profiles of human samples would be desirable.
The authors should provide some representative data of cytokine expression, since NAMPT expression is known to influence cytokine levels.
The study demonstrates the efficacy of FK866 in suppressing the growth of cancer cells in vitro, have you considered testing the inhibitor in mouse models or alternative methods that mimic human pathology?
Author Response
Reviewer 2
In this article, the authors tested the efficacy of the known NAMPT inhibitor, FK866 in reducing the survival of CCA cells. They demonstrated that by preventing the activity of NAMPT, the FK866 inhibitor not only suppresses the growth capacity of cancer cells but also causes changes in mitochondrial metabolism. Furthermore, the experimental results show that a combination of drugs can be a successful strategy for the treatment of tumors resistant to chemotherapy alone. The conclusions are supported by the results and the results obtained are strong. The study has important clinical implications.
Comment: Since several polymorphisms have been found in NAMPT that could cause resistance to treatment, sequencing of RNA profiles of human samples would be desirable.
Response: We really appreciate this comment and agree with the Reviewer. The main goal of our manuscript was to start to evaluate in vitro the potential use of NAMPT as a target for CCA. However, our collaborator and co-author, Dr. Carotenuto, has utilized tumor samples derived from the patient cohort that were analyzed for both RNAseq and Whole Exome Sequencing (WES) profiles. The expression profile of NAMPT obtained by RNAseq data was shown in Figure 1C, while WES analysis conducted on DNA isolated from the same cohort (data not shown) didn't provide any genetic alterations or polymorphism of the NAMPT gene. Nevertheless, the polymorphism analysis definitely warrants further investigations in larger cohorts in the future.
Comment: The authors should provide some representative data of cytokine expression, since NAMPT expression is known to influence cytokine levels.
Response: As the Reviewer suggested, we have now checked the expressions of some cytokines including, IL-6, TGFb, IL-4, and IL-17. Interestingly, IL-6 and TGFb2 showed decreased expression after FK866 treatment. This information is now included in the results section and in Supp Figure 4.
Comment: The study demonstrates the efficacy of FK866 in suppressing the growth of cancer cells in vitro, have you considered testing the inhibitor in mouse models or alternative methods that mimic human pathology?
Response: This is an excellent point that we will explore in the future when funding becomes available. In this first exploration of the topic, the main goal was to have in vitro evidence of the potential use of NAMPT as a target for CCA that will warrant further investigations including, as the reviewers suggest, in vivo rodent models, PDXs, and/or human-derived organoids. Unfortunately, those studies are beyond our reach currently.
